# Role of Lipids and Lipid Metabolism in Prostate Cancer Progression and the Tumor’s Immune Environment

**DOI:** 10.3390/cancers14174293

**Published:** 2022-09-01

**Authors:** Aino Siltari, Heimo Syvälä, Yan-Ru Lou, Yuan Gao, Teemu J. Murtola

**Affiliations:** 1Faculty of Medicine and Health Technology, Tampere University, 33100 Tampere, Finland; 2Department of Pharmacology, Faculty of Medicine, University of Helsinki, 00100 Helsinki, Finland; 3Institute for Regenerative Medicine, Shanghai East Hospital, Tongji University, Shanghai 200070, China; 4Department of Clinical Pharmacy and Pharmaceutical Administration, School of Pharmacy, Fudan University, Shanghai 200437, China; 5TAYS Cancer Center, Tampere University Hospital, 33100 Tampere, Finland

**Keywords:** lipid metabolism, immune response, prostate cancer, cholesterol, T-cells, macrophages

## Abstract

**Simple Summary:**

Cholesterol, lipids, and lipid metabolism are important in prostate cancer. Lipid metabolism interacts with androgens which are of clear importance in prostate cancer. Additionally, lipid metabolism is intimately involved in the interaction between immune and cancer cells. During cancer progression, there are changes in lipid metabolism in both prostate cancer cells and immune cells; furthermore, these cells can interact with each other. Lipids and cholesterol in the circulation also have a role and may prove to be a future target for diagnostic tools and surveillance in prostate cancer.

**Abstract:**

Modulation of lipid metabolism during cancer development and progression is one of the hallmarks of cancer in solid tumors; its importance in prostate cancer (PCa) has been demonstrated in numerous studies. Lipid metabolism is known to interact with androgen receptor signaling, an established driver of PCa progression and castration resistance. Similarly, immune cell infiltration into prostate tissue has been linked with the development and progression of PCa as well as with disturbances in lipid metabolism. Immuno-oncological drugs inhibit immune checkpoints to activate immune cells’ abilities to recognize and destroy cancer cells. These drugs have proved to be successful in treating some solid tumors, but in PCa their efficacy has been poor, with only a small minority of patients demonstrating a treatment response. In this review, we first describe the importance of lipid metabolism in PCa. Second, we collate current information on how modulation of lipid metabolism of cancer cells and the surrounding immune cells may impact the tumor’s immune responses which, in part, may explain the unimpressive results of immune-oncological treatments in PCa.

## 1. Introduction

Prostate cancer (PCa) is the most common malignancy among men and either the second or the third most common cause of cancer death in the developed countries [1]. In 2020, it was estimated that there were 1,414,259 new cases and 375,304 deaths related to PCa [1]. It is likely that the incidence rates for PCa will increase during following decades as the population ages. There are three established risk factors for PCa, i.e., age, family history, and African American race [2]. However, numerous other risk factors have been suggested including genetic and lifestyle related factors. 

While localized prostate cancer can often be managed curatively, treatment of advanced disease is based on androgen deprivation since the growth of PCa initially relies on androgens. Unfortunately, PCa eventually develops castration resistance, i.e., its ability to grow and progress despite a low androgen level. 

Lipid metabolism is known to be disturbed in cancer cells [3]. In addition to being used as an energy source, lipids are used for cell membrane biosynthesis, signal transduction, intracellular trafficking, cell polarization, and migration, features which are also important for cancer development and progression. Thus, changes in the regulation of lipid metabolism are one of the hallmarks of cancer [4]. 

The growth and progression of PCa depend on androgen receptor (AR) signaling, which is the target of established oncological treatments of advanced PCa. Activation of AR targets several genes in the lipid metabolic pathway [5,6,7]. Concordantly, it is well established that there are changes occurring in lipid metabolism during the development of PCa [5,8].

Intraprostatic inflammation is another trait associated with PCa and which can be affected by androgens. In humans, androgen deprivation therapy leads to increased infiltration by immune cells into the prostate [9]. In addition, an acute bacterial infection in the prostate has been associated with an increased PCa risk [10]. More recently, research interest has also focused on the possible association of the direct and indirect impact of the human microbiota, e.g., the urinary microbiome, on prostate tissue, which might influence the PCa risk [11].

Inflammation and immune cell infiltration may have dual roles in the microenvironment surrounding cancer cells [12]. While on one hand, inflammation can promote tumor development and progression, on the other hand, immune cells can identify and destroy dysfunctional cells such as cancer cells. Understanding the control and modulation of immunological responses in the tumor microenvironment during the development and progression of cancer is crucial to understanding how to promote tumor suppression and decrease tumor promoting actions of the immune system. 

Cancers can be roughly categorized into immunologically hot or cold types based on their immune responses and immunological activity [13]. In immunologically hot tumors, immune cell infiltration is high, and the immune cells are relatively active. In immunologically cold tumors, infiltration by immune cells is reduced, and the cells are not as active as in hot tumors. PCa is considered an immunologically cold tumor [9] despite intraprostatic inflammation being very common. Therefore, PCa cells likely are efficient in avoiding immune control. 

Immune-oncological (IO) treatments targeted to activate cytotoxic T-cells have mostly failed in PCa treatment; only a minority of patients have exhibited good treatment responses [14]. The mechanisms behind low response rates are not well understood; however, the tumor mutation burden, which predicts the IO treatment response in some other cancer types, is relatively low in PCa in comparison to the cancers that respond better to IO treatments [13].

Recently, research interest has been focused on the possible link between the regulation of lipid metabolism and immune responses. It remains unclear to what degree the altered lipid metabolism in PCa cells modulates the regulation of the immune response, and whether lipid metabolism is reciprocally altered by immune responses. 

This review assesses the current knowledge about how lipid metabolism interacts with the immune environment around the PCa tumor cells. First, we describe how lipid metabolism is involved in PCa progression. Second, we review current knowledge on how lipid metabolism in surrounding immune cells is altered in PCa and other solid tumors, and how this may impact the regulation of tumor immune responses. Finally, we review how these may be affected by the circulating lipids and cholesterol.

## 2. Lipid Metabolism Is Regulated by SREBP and Interacts with AR Signaling in Prostate Cancer

Lipid metabolism is upregulated in PCa, especially in the context of castration resistance. Lounis et al. [15] conducted an RNAseq analysis of de novo PCa cells and noted that lipid metabolism increased during the development of castration resistance. Furthermore, based on the RNAseq data, de novo lipid synthesis was even more upregulated in castration resistant cells in comparison to enzalutamide resistant cells. When hormone sensitive and castration resistant cells were treated with a combination of enzalutamide and a SCD1 inhibitor (inhibitor of lipogenesis), cell death was increased in both cell lines when compared to enzalutamide alone.

In PCa cells, AR modulates the expression of many important genes in the lipid metabolism pathways (Figure 1). Sterol regulatory element-binding proteins (SREBPs; 1a, 1c, and 2) and fatty acid synthase (FASN) are especially upregulated in an AR-dependent manner [5,6]. AR activates the expression of SREBP cleavage-activating protein (SCAP) which is needed for the activation of SREBP. The AR also inhibits liver X receptors (LXR) which have several functions in the regulation of lipid metabolism [7]. Protein levels of SREBP1 were increased in human PCa specimens as compared to non-cancerous tissue [16]. In addition, in PCa cells and xenografts, SREBSs were upregulated, especially after castration resistance [16]. Huang et al. [17] showed that SREBP1 expression in PCa was positively associated with the tumor’s Gleason grade. In the same study, overexpression of SREBP1 increased the expression of AR, FASN, and lipid accumulation in the cells, while SREBP1 downregulation lowered these parameters. SREBP1 was also linked to development of castration resistance. 

SREBPs act as transcription factors which bind to sterol regulatory element or E-boxes in the promoter area and activate the expression of lipid metabolism-related genes such as low-density lipoprotein (LDL) receptor and HMG CoA synthase and reductase genes [18]. SREBPs have three different isoforms: SREBP 1a, 1c, and 2. Inactive SREBPs are membrane bound proteins which are activated after cleavage from the endoplasmic reticulum membrane in a two-step process in the Golgi apparatus. Membrane-free SREBP is transported to the nucleus where it activates the expression of lipid metabolism-related genes. Cleaved SCAP is transported back to the membrane where it forms a complex with another SREBP molecule. The SREBP activity is mainly controlled by the intracellular sterol concentration. However, other sterol-independent SREBP activating molecules have been suggested such as TNF-α and mTORC1 [18,19,20].

Lipid metabolism is also affected by a host of external influences and other metabolic disorders which are too numerous to cover in this article. A review on that topic has been published elsewhere [21].

Because of their crucial role in the regulation of lipid metabolism, SREBPs can be thought of as master regulators of fatty acid and cholesterol synthesis since not only do they induce the activation of FASN, but they also upregulate cholesterol metabolism by inducing the mevalonate pathway (Figure 1). In particular, the SREBP2 isoform induces cholesterol metabolism and promotes the intracellular accumulation of cholesterol [22] while SREBP1 is considered to activate fatty acid synthesis. Furthermore, SREBPs act as oxygen sensors which help cells to adapt to a hypoxic microenvironment, another factor commonly encountered in solid tumors, in which lipid metabolism is upregulated [23]. It is also noteworthy that SREBP1 increases the expression of the AR; thus, the AR–SREBP1-axis forms a self-regulating loop to ensure a continuous gene expression of transcription factors [17]. 

The importance of SREBP activity for prostate cancer growth is now well demonstrated. Preclinical studies have shown that inhibition of SREBP signaling using fatostatin, a compound which blocks the activity of all forms of SREBP by inhibiting the binding of the SCAP protein to SREBP, suppressing cell proliferation in AR-positive prostate cancer cells, decreasing AR-positive prostate tumor growth, and lowering blood PSA in mice. When combined with docetaxel, fatostatin decreased the proliferation and induced cellular apoptosis in AR-negative cells especially if the cells harbored the p53 mutation [24,25]. Furthermore, treatment with fatostatin decreased tumor growth and the formation of lymph node metastases in PTEN knockout mice, a mouse model of prostate cancer [26].

FASN likely has a role in PCa progression. In human samples, the expressions of the FASN and androgen receptor variant 7 (AR-V7) were found in many metastatic tumors, highlighting their importance in advanced PCa [27]. The inhibition of FASN in different castration resistant cell lines decreased tumor growth, increased apoptosis, altered the lipidome, decreased lipid accumulation, and decreased the expression of the AR and AR-V7 [28]. These AR activated pathways are linked to endoplasmic reticulum (ER) stress which means there are disturbances in clearing damaged folded proteins and a malfunction in the regulation of apoptosis [27,28]. Thus, inhibition of lipid metabolism can decrease ER stress by impacting the activity of AR-related pathways. 

Thus, in summary, a modulation of lipid metabolism in PCa has been shown to be important especially during cancer progression and the development of drug resistance. The AR regulates lipid metabolism especially by inducing SCAP expression, mandatory conjugates for SREBP activation. SREBPs in turn augment the expression of the AR. Thus, inhibition of SREBPs and FASN can be considered as potential targets for future PCa therapies. 

## 3. Lipidomic Changes during PCa Progression and Castration Resistance

Burch et al. [29] investigated the intracellular lipidome and metabolomics, focusing especially on lipid pathways in two non-malignant cell lines, one prostate adenocarcinoma, and two metastatic prostate cancer cell lines; all were AR positive. Compared to non-malignant and non-metastatic cells, many classes of lipids were upregulated in metastatic cell lines, including phosphatidylcholines, phosphatidylethanolamines, and glycophosphoinositols. Further, both the enzymes and pathways responsible for de novo lipid synthesis were upregulated in metastatic cells, supporting the concept that especially in advanced PCa, lipids are produced de novo rather than being taken up from the circulation. 

In another study, the lipidome and activity of lipid pathways were characterized in non-cancerous prostate cells and hormone-sensitive, castration resistant, and docetaxel resistant PCa cell lines [30]. While glycerophospholipid metabolism was the most enriched pathway in PCa cell lines compared to normal prostate cells, sphingolipid metabolism, sphingolipid signaling pathway, ferroptosis, necroptosis, and phospholipase D signaling pathways were also enhanced in PCa cells. When docetaxel resistant cell lines were compared to non-cancerous cells, altogether 21 different lipid species were altered. Furthermore, when drug-resistant cell lines were compared to their parent cell line (in this case PC3 and DU145 PCa cell lines), it was found that levels of phosphatidylcholine, oxidized lipid species, phosphatidyl-ethanolamine, and sphingomyelin were upregulated [30]. 

Li et al. [31] investigated lipidomic changes in prostate tissue samples from PCa patients. Many lipid metabolism-related pathways were dysfunctional in PCa tissue. Cholesterol esters were more prominently accumulated in PCa tissue as compared to non-cancerous tissue; these compounds were also associated with the progression and development of metastatic disease. The recent study by Butler et al. [32] also evaluated the lipidome from human tissue samples collected from prostate cancer patients during prostatectomy. The analysis included tumor samples and matched non-malignant samples from 21 patients and unmatched PCa samples from 47 patients. Furthermore, prostate tissue from unmatched subjects was cultured ex vivo with or without enzalutamide. When matched specimens were compared, it was evident that tumors had a specific lipidome; for instance, the total phospholipid content differed between tumor and benign tissue. With respect to the individual lipid species, phosphatidylcholines were the most abundant lipids in both tumor and normal tissue, but highest in tumor samples. The amount of lipids with one or two monounsaturated fatty acyl chains was upregulated in tumor samples. Furthermore, ex vivo culturing with enzalutamide changed the cellular lipidome in comparison to cells cultured without the drug. The authors noted that ex vivo culturing of cancer tissues alone had exerted only a minimal impact on the lipidome, thus human tissue culture might be reliable for testing of lipidomic changes in response to external influences, e.g., drug treatments.

## 4. Lipidome as Biomarker in Prostate Cancer

Lipidomic changes in liquid biopsies using blood or urine samples have been investigated among PCa patients for the purpose of finding novel biomarkers for PCa detection. 

In 2000, Filella et al. [33] noticed in a small patient group of PCa patients and healthy controls that the balance between type 1 cytokines, IL-2 and INF-γ, and type 2 cytokines, IL-4 and IL-10, was different as compared to healthy donors’ blood. More recently, Lin et al. [34] examined if there was a correlation between levels of cytokines and the lipidome in advanced PCa patients’ blood samples. In general, they concluded that the serum lipidome was a more reliable prognostic marker than cytokines in these patients. In particular, a high serum concentration of sphingolipids at baseline was associated with a poor prognosis. The increased concentration of a set of different cytokines was associated with an elevated level of ceramides. Ceramides are the backbone and basic structural unit in all sphingolipids. 

Zhou et al. [35] were able to distinguish PCa patients from healthy patients based on the serum lipidome using principal component and hierarchical clustering analyses. They also suggested that phosphatidylethanolamines, ether-linked phosphatidylethanolamines, and ether-linked phosphatidylcholines could be used as markers for PCa diagnosis. Another study investigated changes in the serum lipidome between PCa patients and healthy donors [36]. They were also able to separate PCa patients from healthy controls based on a principal component analysis of the serum lipidome. Phosphatidylcholines (39:6) and fatty acids (22:3) were the best candidate lipids for classifying samples into PCa and healthy patients.

In urine and prostate tissue samples, a high ratio of phosphatidylcholines and lysophosphatidylcholines was associated with PCa compared to samples from benign prostate hyperplasia patients in a Japanese trial [37]. Lin et al. [38] investigated the association between the plasma lipidome and clinical outcomes in castration resistant PCa patients. They found an association between the amount of sphingolipids and poor prognosis. Furthermore, a three-lipid species signature of ceramides (18:1/24:1), sphingomyelins (18:2/16:0), and phosphatidylcholines (16:0/16:0) was associated with a poorer prognosis among these patients. The prognostic value of the three-lipid species signature was also evaluated in a recent study in metastatic castration resistant patients [39]. Overall survival was found to be poorer in patients with the signature as compared to those not expressing it. In the same study, lipid species of acylcarnitines and ceramides were associated with a poor outcome in all stages of PCa. Recently, glycerophospholipids and glycosphingolipids have also been suggested as potential biomarkers in the blood lipidome analysis for PCa detection [40,41].

Taken together, the lipidome profile is altered in tissue samples and liquid biopsy specimens of PCa patients as well as PCa cells compared to non-cancerous cells. The lipidome seems to be constantly modulated during cancer progression and castration resistance. Unfortunately, it remains unclear whether a lipidomic change is the cause or consequence of cancer progression. Nevertheless, in the future, either the lipidome or specific lipid species might prove to be prognostic for PCa surveillance and could possibly be used as a diagnostic tool for PCa diagnosis. 

## 5. Cholesterol Metabolism in PCa

Distinct from lipid metabolism, cholesterol metabolism also has an important role in prostate tissue and in PCa progression; while benign prostate epithelial cells synthesize cholesterol, it seems that PCa cells accumulate even more cholesterol. The abundance of cholesterol in PCa cells is due to increased cholesterol production de novo and elevated uptake from the circulation [42,43]. In addition to its role as the central component of cell membranes, cholesterol is a substrate in the biosynthesis of all steroid hormones, including androgens. Thus, many investigators have been interested in clarifying its role during PCa progression. Cells can obtain cholesterol from the serum, which in turn is influenced by the diet and cholesterol production in the liver. Furthermore, PCa cells produce cholesterol de novo through the mevalonate pathway. Since a too high concentration of cholesterol is toxic for the cells, excess cholesterol is esterified and either stored in lipid rafts along with other lipids or refluxed from the cells using specific transport proteins, ATP-binding cassette transporter A1 or G1 (ABCA1 or ABCG1). 

In PCa, it has been shown that in comparison to normal cells, cancer cells are able to produce more cholesterol; the cholesterol producing pathways and cholesterol uptake are upregulated whereas the cholesterol efflux is downregulated. These alterations are known to occur especially during PCa progression [5]. Cholesterol and LDL are important in PCa cell metabolism compared to normal prostate epithelial cells [44,45]. Statins, i.e., cholesterol lowering drugs which inhibit the mevalonate pathway by inhibiting HMG-CoA reductase (HMGCR), have been shown to decrease PCa cell growth, invasion, and migration by inducing apoptosis and arresting cell growth [44,45,46,47,48]. Statins do not influence the expression of cyclo-oxygenase 2 enzymes [46], thus they do not affect inflammation directly. It was reported that treatment with simvastatin also decreased tumor growth in nude mice [48]. Furthermore, an increased expression of HMGCR has been associated with a poor prognosis in PCa patients [49]. The expression of HMGCR is increased in enzalutamide-resistant PCa cells [50]. The knockdown of this enzyme re-sensitized enzalutamide-resistant cells so that they responded to enzalutamide, pointing to the role of HMGCR in enzalutamide resistance. Furthermore, the combination of simvastatin with enzalutamide decreased PCa cell growth more than either drug on its own in both in vitro and in vivo models of enzalutamide resistance. The authors also showed that simvastatin alone and in combination with enzalutamide decreased AR protein expression in enzalutamide resistant cells. 

Cholesterol metabolism in PCa cells is affected by the interaction with the microenvironment. Co-culturing of PCa cells and cancer-associated fibroblasts (CAFs) in a three-dimensional (3D) culture, gene expression analysis of PCa cells revealed that cholesterol and steroid biosynthesis pathways were upregulated as compared to the state where PCa cells were cultured without the presence of CAFs [51]. Especially 3-hydroxy-3-methylglutaryl-coenzyme A synthase 2 (HMGCS2) and aldo-keto reductase family 1 member C3 (AKR1C3) genes were upregulated, even when PCa cells were cultured with CAF culturing media. CAFs were shown to secrete a high amount of proinflammatory cytokines and chemokines to the medium. When co-cultured cells were treated with simvastatin, the growth of the spheroids was decreased. The protein expressions of both HMGCS2 and AKR1C3 were higher in PCa epithelial cells as compared to benign tissue in human prostate tissue samples. Furthermore, AKR1C3 protein expression was correlated with the tumor’s Gleason grade.

In summary, cholesterol is crucial for prostate cancer cells, and its production is upregulated during the development of castration resistance. In addition, cholesterol production likely has an important role in enzalutamide resistance. Cholesterol metabolism in PCa cells is affected by the microenvironment surrounding the tumor.

## 6. Impact of Inhibition of the Cholesterol-Producing Mevalonate Pathway with Statins

Even though statin treatment in preclinical models seemed to inhibit the growth of tumor cells, one study in experimental animals administered a low-dose of statin (50 nM) and showed opposite results; after low-dose statin treatment, PCa tumors in mice were growing even better than in the untreated mice [52]. These investigators tested the low-dose statin setting in obese mice, castrated mice, and in human cell xenografts but their conclusions remained the same; low-dose statin treatment exerted no anticancer actions. These results may be explained by a trial in which fluvastatin treatment in different PCa cells activated SREBP2 expression via a feedback mechanism, which in turn increased HMGCR expression. Thus, statins in low doses are not effective enough to inhibit the increased expression of the enzyme [49]. The combination of statin and SREBP2 inhibitor and SREBP2 silencing decreased fluvastatin’s IC(50) values, thus sensitizing the cells to fluvastatin treatment [49]. Wang et al. reported that statins do not reliably lower serum cholesterol in mice [53]. 

It was demonstrated that androgen depletion enhanced the transition of bone marrow stromal cells into adipocytes [54]. Adipocyte-secreted factors, including leptin, in turn stimulated the PCa cell cycle progression and cell proliferation through Stat3 activation and statins have been postulated to suppress this transition in vitro and in vivo [53].

So far, only a few clinical trials investigating the impact of statin use on prostate cancer progression have been conducted. We performed a clinical trial assessing the impact of high-dose (80 mg daily) atorvastatin administration on prostate cancer tissue and blood markers before prostatectomy [55]. In the trial, men used either atorvastatin or placebo for a median of 26 days before prostatectomy. When atorvastatin was used for longer than 27 days, prostate tissue Ki-67 proliferation index was decreased in the atorvastatin group as compared to placebo. In addition, the PSA level decreased among patients with high-grade PCa in comparison to placebo. Furthermore, atorvastatin was measurable in the prostate tissue; thus, the dose had been high enough for tissue penetration [56]. Atorvastatin treatment also changed the subjects’ lipidome and androgen profile in prostate tissue [57,58]. Prostate tissue microRNAs were also evaluated [59] but no clear correlations were observed between microRNAs and tumor clinical characteristics in either of the study arms. Many of the altered microRNAs were concluded to be participating in several pathways which are important during cancer development and progression.

Recently Jeong et al. [60] investigated the role of low-dose therapy with atorvastatin after prostatectomy. Men were using 20 mg of atorvastatin or placebo daily for one year after surgery. The use of low-dose atorvastatin had no impact on biochemical recurrence within five years after surgery. Based on pre-clinical and epidemiological findings, it seems that cholesterol metabolism is one of the key regulators during the progression of advanced PCa, and therefore we are currently conducting a phase III randomized clinical trial on the impact of adjuvant high-dose atorvastatin treatment with advanced prostate cancer patients on long-term androgen deprivation therapy (ADT) [61]. The primary endpoint of this trial is the time until the appearance of castration resistance from the start of ADT; thus, our main objective is to determine whether the use of statin in combination with ADT will be able to delay the formation of castration resistance, the last stage of PCa. Overall survival will be a secondary endpoint of this trial.

These trials once again underline the importance of understanding cholesterol metabolism as a whole during PCa progression and indicate that perhaps statins need to be used in high doses in order to elicit beneficial responses in PCa.

## 7. Fatty Acid Metabolism in PCa

Fatty acid metabolism is important for tumor development and progression in many ways, such as increasing the synthesis and storage of fatty acids and decreasing lipotoxicity and ferroptosis in tumors [62]. Maintaining optimal homeostasis of fatty acid species (such as the ratio of monounsaturated to saturated and the ratio of monounsaturated to polyunsaturated fatty acids) is essential to avoid lipotoxicity and ferroptosis.

Ferroptosis is a recently identified new form of programmed cell death. Ferroptosis is caused by the accumulation of iron-dependent lipid peroxides in the cells. Lipid metabolism is the main regulator of this event. The role of ferroptosis in cancer development and progression is under investigation. Ferroptosis might also impact the immune responses against tumors. Thus, targeting the upregulation of ferroptosis in tumor cells might be a good target for cancer treatment, especially in cancers with high lipid metabolism. This topic has been recently reviewed [63,64]. Here we focus on research related to ferroptosis and PCa.

An evaluation of the RNA-Seq in prostate cancer tumors compared to noncancerous prostate tissue showed that the gene panel of ferroptosis-related genes was differently expressed in tumors as compared to healthy tissue [65]. Based on a subgroup of seven ferroptosis-related genes, a risk score for the prognostic model was created and patients were divided into two groups based on their calculated risk scores (low and high risk). According to the survival analysis, the high-risk score group had a lower survival rate than patients in the low-risk group. 

While it is known that AR activation promotes lipid accumulation in prostate cancer cells, intracellular lipid levels were increased even after treatment with enzalutamide, an AR signaling inhibitor [66]. Especially, modulation of the polyunsaturated fatty acid (PUFA) content by enzalutamide treatment was associated with increased lipid peroxidation and led to hypersensitivity to ferroptosis. In PCa cells highly sensitive to iron toxicity (VCaP, LNCaP, and TRAMP-C2), a high iron concentration additionally evoked protein damage [67]. Thus, apart from the lipid modifications, iron induced cell death by increasing ferroptosis, strengthening the efficacy of anti-androgen therapy in PCa models. Furthermore, erastin, a classical inducer of ferroptosis, suppressed the transcriptional activities of both the full length and the splice variant of AR in castration resistant prostate cancer cells [68]. Treatment with erastin also enhanced efficacy of docetaxel treatment in a castration resistant PCa model.

It appears that cancer cells can downregulate ferroptosis via a SREBP1 mediated pathway: in tumors where PI3K–Akt–mTOR signaling is active, SREBP1 is activated, and this protein suppresses ferroptosis [69]. When SREBP1 was inhibited in the cell model, the cells became sensitized to ferroptosis. 

Castration levels of testosterone inhibited the growth of testosterone sensitive PCa cells by inducing ferroptosis which led to increased immune cell infiltration into the tumor site [62,70]. In prostate cancer cells and mouse models, ferroptosis inducers were able to decrease cell and tumor growth by inducing ferroptosis [71]. The decline in tumor growth in vivo was even more evident when ferroptosis inducers were given together with enzalutamide or abiraterone, drugs used to treat advanced PCa. 

Increasing ferroptosis by modulation of lipid metabolism in PCa might be a novel target for treatment to enhance the impact of the traditional therapies. The impact of the induction of ferroptosis in the immune environment of the PCa tumor needs further clarification. 

AR activation can not only induce energy production via glycolysis but also upregulate de novo fatty acid synthesis and fatty acid uptake and oxidation [72,73]. Lipids can be utilized as energy due to the β-oxidation of fatty acids in the mitochondria. Long-chain fatty acids require being transported into the carnitine shuttle to allow them to enter mitochondria for β-oxidation and in prostate cancer, both mechanisms are modulated. For instance, overexpression of a key enzyme of β-oxidation, Δ2-Enoyl-CoA Delta Isomerase 1 (ECI1), increased PCa cell growth. On the other hand, inhibition of this enzyme decreased the growth of the cancer cells [74]. In clinical prostate samples, overexpression of ECI1 was associated with the risk of biochemical recurrence [74]. The importance of carnitine shuttle upregulation in PCa cells has been demonstrated [75]. In a recent mass spectrometry imaging analysis, increased expression of compartments of carnitine shuttle was also revealed in the pairwise analysis of prostate cancer, benign prostate, and stroma tissue clinical samples [76]. 

In castration resistance, PCa cells typically display over-expression of AR-regulated metabolic genes compared to androgen-sensitive tissues or cells. Thus, interest has been focused on whether lipid metabolic processes could be targeted as a novel treatment for advanced PCa [77,78]. Several recent studies have investigated the efficacy of targeting lipid metabolic enzymes as either a monotherapy or in combination with AR-signaling inhibitors in castration resistant PCa models. For example, inhibition of carnitine palmitoyltransferase I (CPT1) or FAS enhanced the sensitivity to androgen receptor antagonists in preclinical models [27,79,80]. The crosstalk between AR expression and signaling, and the modulation of fatty acid metabolism may provide a future target for re-sensitizing treatment resistant PCa cells.

In antiandrogen resistance, there is an increased number of glycerophospholipid species with longer and more unsaturated fatty acyl chains detected in PCa cells [68]. Furthermore, the gene encoding 2,4 dienoyl-CoA reductase (DECR1), an enzyme catalyzing the rate-limiting step in polyunsaturated fatty acyl-CoA oxidation, is overexpressed in castration resistant tumors as compared to primary tumors [81]. Overexpression of DECR1 is also associated with a shorter relapse-free time and overall survival. Knockdown of DECR1 evoked a decrease in the proliferation, migration, and treatment resistance in PCa cells as compared to control cells. ER stress and ferroptosis were also increased after DECR1 knockdown [82]. 

In summary, fatty acid metabolism appears to be another crucial metabolic factor in castration resistance and may provide a target for reversing of resistance to antiandrogen resistance. Fatty acid metabolism appears to be linked with ferroptosis in PCa.

## 8. Changes in the Lipidome and Cholesterol Metabolism in Cancer-Associated Immune Cells

The metabolism and supply of lipids in both prostate cancer cells and immune cells of the tumor microenvironment may be altered at multiple levels. First, the disturbed lipid metabolism in cancer cells may affect the surrounding immune cells. Second, changes in lipid metabolism in the immune cells may affect their activity and reciprocally also affect cancer cells. Third, the supply of lipids from the serum may promote changes in both immune and cancerous cells.

In a mouse model of melanoma, gene enrichment analysis of tumors revealed that lipid associated metabolic pathways were enriched in Treg cells in the tumor as compared to cells from peripheral tissue [83]. A further analysis revealed that Tregs needed SREBP/SCAP activity in order to perform their active functions in the tumor site. Tumor growth was also decreased in mice with SCAP deleted Treg cells and these mice were also more sensitive to anti-PD1 therapy. The authors concluded that metabolic programming of the fatty acid and mevalonate pathways by SREBP in Treg cells, especially the pathway related to protein geranylgeranylation, was able to impact Treg function in tumors. In addition, Treg cells suppressed the secretion of interferon-γ (IFN-γ) in CD8+ T cells. It is recognized that IFN-γ secretion can block the activation of SREBP1-mediated fatty acid synthesis in immunosuppressive (M2-like) tumor-associated macrophages (TAMs), thus Treg cells were able to activate TAMs by modulating the secretion of IFN-γ [84]. Furthermore, SREBP1 inhibition augmented the efficacy of immune checkpoint blockade, suggesting that the targeting of Treg cells, e.g., by altering their modulation of lipid metabolism in TAMs, could be a way of improving cancer immunotherapy. In addition, a high production of free fatty acids evoked by N-cadherin in a mouse model promoted the formation of Treg cells and increased immune suppression [85]. Recently, it was shown that AR activity in T-cells could suppress CD8+ T-cell activity by reducing IFN-γ secretion and inducing T-cell exhaustion and resistance to IO treatment [86].

Recently the role of cholesterol in immune cell regulation in solid tumors has become a focus of research interest. Ma et al. showed that an increased cholesterol content in CD8+ T-cells and in the tumor microenvironment increased the expression of several immune checkpoints, PD-1, 2B4, TIM-3, and LAG-3, and led to T-cell exhaustion [87]. Thus, by reducing the cholesterol concentration in CD8+ cells, the researchers were able to restore the cytotoxic function of T-cells. The mechanism behind how cholesterol induced CD8+ exhaustion and interrupted T-cell metabolism was linked to an activation of ER stress-response genes. Since the cholesterol content also regulated the IL-9 producing subtype of CD8+ cells (Tc9 cells), by reducing the cholesterol level, it was possible to increase the antitumor effect of Tc9 cells [88]. Tc9 cells also express cholesterol metabolizing enzymes and cholesterol efflux is high; thus, intracellular cholesterol levels were low in these cells as compared to Tc1 cells (a subtype of CD8+ cells secreting IFN-γ) [88]. There is some indication that the cholesterol content of the plasma membrane of the tumor cells can create a mechanical barrier against cytotoxic T-cells by softening the cell membrane. Cell stiffening by cholesterol depletion may increase the impact of T-cell cytotoxicity [89].

Many cholesterol metabolism related enzymes have an impact on the activity of CD8+ T-cells. Yang et al. [90] showed that inhibition of acetyl-CoA cholesterol acyltransferase (ACAT1) increased the proliferation of CD8+ T-cells, i.e., cells with ACAT1 knockdown were more active as compared to wildtype T-cells in their ability to depress melanoma cell growth in a mouse model. Inhibition of cholesterol acyltransferase in liver carcinoma cells increased T-cell activity by restoring the activity of CD8+ exhausted T-cells and increasing the amount of CD8+ T-cells [91]. Knockdown of protein convertase subtilisin/kexin 9 (PCSK9) resulted in less extensive xenograft growth and longer survival times in breast cancer, colon cancer, and melanoma xenografts [92]. PCSK9 reduces cholesterol metabolism by down-regulating the numbers of low-density lipoprotein receptors (LDLR) on the cell surface. Although inhibition of this enzyme increases T-cell infiltration in the tumors, this effect is not dependent on the presence of LDLR. PCSK9 inhibition increased expression of MHC1 proteins on the surface of cancer cells which appeared to increase T-cell infiltration. Importantly, genetically proxied inhibition of PCSK9 was associated with a lowered risk of PCa [93]. Moreover, the expressions of both IFN-γ and IL-2 were suppressed while that of IL-6 was up-regulated in CD4+ T-cells collected from male C57Bl/6 mice which had been exposed to media conditioned with macrophages grown in sera from obese humans. Furthermore, exposure to conditioned media of obesity-modified CD4+ T-cells increased the expression of epithelial–mesenchymal transition markers in PCa cells, a feature which elevated their invasive and migratory properties [94]. Ezetimibe, a drug which inhibits the permeability of cholesterol to be taken up from the intestine, not only reduced serum cholesterol levels, but also inhibited mTORC2 signaling in CD8+ T-cells, increased infiltration of CD8+ T-cells into prostate tumors, and enhanced CD8+ memory lymphocytes with a central memory phenotype [53].

Changes in lipid metabolism of immune cells can induce treatment resistance in prostate cancer cells. In a pre-clinical trial exploiting both mice and cell models, El-Kenawi et al. [95] were able to show that macrophages were associated with cholesterol transport and androgen synthesis in PCa, i.e., the macrophages were regulating AR nuclear translocation, thus contributing to the formation of enzalutamide resistance. Furthermore, tumor associated anti-inflammatory M2 macrophages were shown to contain a high concentration of intracellular cholesterol. When co-cultured with PCa epithelial cells, M2 TAMs were able to transfer cholesterol to the epithelial cells, in other words, they were acting as a cholesterol source for the cancer cells. In addition, cholesterol metabolism seems to be an important regulator of macrophage polarization; in the ABCG1 knockdown (transporter for cholesterol efflux) mouse strain, macrophage polarization switched from the tumor-promoting M2 to the anti-tumor-promoting M1 phenotype [96]. Similar switches between M1/M2 phenotypes were seen after inhibition of ABCG1 in human macrophages. Stimulation of human macrophages with HDL also downregulated the polarization to the M1 phenotype; however, it had no impact on M2 polarization [97]. A high HDL content decreased the amount of caveolin-1 on the cell surface of macrophages, which inhibited their polarization to the M1 phenotype. Exposure to cancer cell derived IL-1β enhanced the expression of the scavenger receptor, marco, on a subset of macrophages [98]. These macrophages were associated with PCa progression and a shorter disease-free survival. Marco was shown to regulate the accumulation of lipids into the macrophages, e.g., its inhibition decreased tumor growth and invasiveness in a mouse model. Moreover, provision of a high-fat diet in a mouse model increased the number of lipid-loaded TAMs. In a mouse model of prostate cancer, feeding with a high-fat diet increased the number of myeloid-derived suppressor cells as well as elevating the ratio of M2/M1 macrophages [99]. Treatment with celecoxib, a cyclooxygenase 2 inhibitor and an anti-inflammatory drug, decreased tumor growth, as well as reducing the number of myeloid-derived suppressor cells and the M2/M1 ratio. The regulation of responses to a high-fat diet in terms of inflammation and tumor growth was related to IL6/pSTAT3 signaling also in tissue specimens from obese prostate cancer patients.

In castrated and eugonadal mice fed with an omega-3-enriched diet, the growth of TRAM-C2 cell xenografts was decreased in comparison to mice fed with an omega-6-enriched diet [100]. An omega-3 diet increased Th1-related cytokine production in tumors and increased eosinophil recruitment.

A summary of studies in this section is collected in Table 1; it demonstrates that the lipid metabolism of immune cells becomes disturbed when the cells interact with cancer cells. Tumor associated immune cells may provide cholesterol to cancer cells, thus promoting their growth. Enzymes in lipid and cholesterol pathways affect the polarization of surrounding immune cells, affecting their antitumor properties.

## 9. Pharmacological Interventions Targeting Lipid Metabolism Affect Immune Cells

Pharmacological interventions targeting lipid metabolism have demonstrated effects on immune cells in the tumor microenvironment in cases of prostate cancer. In the ESTO1 trial, atorvastatin use before prostatectomy did not significantly change the intraprostatic inflammation score compared to placebo when all subjects were analyzed together. However, when the analysis excluded patients who dropped out during the trial (did not use the study drugs), the intraprostatic inflammation score was decreased among statin users in men with high-grade PCa [55]. 

In the REduction by DUtasteride of prostate Cancer Events (REDUCE) trial, a high level of high-density lipoprotein (HDL) in the serum was associated with a lower risk for acute inflammation in prostate tissue, while serum levels of total cholesterol, LDL, and triglycerides had no impact [100]. On the other hand, statin use was associated with a decreased level of chronic inflammation in prostate tissue [101]. 

The Prostate Cancer Prevention Trial (PCPT) randomized men with no known history of prostate cancer to either finasteride, a 5α-reductase inhibitor of conversion of testosterone to dihydrotestosterone, or placebo. In the placebo arm, chronic inflammation in prostate biopsies was associated with the risk of high-grade PCa [102]. In the finasteride arm, intraprostatic inflammation was even more abundant but no longer associated with the overall risk of PCa, nor of high-grade tumors [103]. Furthermore, the use of statins was associated with a lower amount of the tissue marker (CD68) of macrophages as compared to non-users [104]. Thus, statin use may block the supply of cholesterol to cancer cells from macrophages. 

Furthermore, a positive correlation between LDL levels and pro-inflammatory (M1) macrophages and a negative correlation between LDL and anti-inflammatory (M2) macrophages was shown in human adipose tissue collected from healthy kidney donors [105]. Treatment with fluvastatin decreased macrophage polarization towards the pro-inflammatory M1 phenotype and increased the anti-inflammatory M2 polarization. It is unknown whether statins would elicit similar responses in the macrophages located in the tumor microenvironment. The association between macrophage polarization, androgen metabolism, and cholesterol metabolism needs to be clarified, especially in human samples.

In addition, exposure to the tyrosine kinase inhibitor, ESK981, decreased tumor growth in prostate cancer cell lines and mouse xenografts by targeting 1-phosphatidylinositol-3-phosphate 5-kinase (PIKfyve), a lipid kinase [106]. Inhibition of lipid oxidation with ranolazine in the TRAMPC1 mouse model resulted in decreased expression of the immune checkpoint block protein Tim3 in CD8+ cells, increased the content of macrophages in the tumor site, and decreased the number of immunosuppressive monocytes in the blood [107].

Treatment with ezetimibe decreased the tumor growth of several cancer cells, including prostate cancer cells, in mice [53]. This was related to changes in immune responses; mammalian target of rapamycin (mTOR) signaling was decreased in lymphocytes affecting the AKT signaling pathway. CD8+ immune cell infiltration also increased in the tumor site after drug treatment. Additionally, the level of the proliferation marker, the Ki-67 index, was evaluated in prostate tumor samples in a window-of-opportunity clinical trial, where men were treated with ezetimibe and simvastatin for two to six weeks before radical prostatectomy. The Ki-67 index after the intervention decreased as compared to the pretreatment state in low-grade tumors, but not in high-grade tumor samples. Based on RNA-seq results in a small subgroup, it seemed that the numbers of CD8+ cells and M1 macrophages had increased after treatment with cholesterol-lowering drugs. 

## 10. Periprostatic Adipose Tissue

The prostate is surrounded by white adipose tissue called the periprostatic adipose tissue (PPAT). Prostate and PPAT are separated by the prostate capsule, however, the prostate vasculature crosses the PPAT [108]. As PPAT consists of white adipose tissue, its purpose is thought to be energy storage. However, PPAT also secretes adipokines, cytokines, chemokines, and growth factors, which in part might stimulate the development and progression of PCa. It has been speculated that contact with PPAT may enhance PCa progression in tumors with extraprostatic extension, leading to decreased biochemical recurrence-free survival after prostatectomy as compared to tumors not invading outside the prostate capsule [109].

PPAT serves as a source of fatty acids for tumor cells. There is some evidence that PPAT can secrete the adipokine, leptin, which might stimulate PCa progression [110,111,112]. However, larger studies will be needed to confirm the role of this adipokine in PCa. Furthermore, PPAT secretes cytokines, such as IL-6 and TNFα; these are chemokines which are known to be involved in the development and progression of PCa [112,113,114]. 

## 11. Conclusions

Cholesterol, lipids, and lipid metabolism play key roles in the progression of PCa. While lipid metabolism interacts with AR signaling, in addition it is closely involved in the regulation of the immune system in the tumor microenvironment. Disturbances in the regulation and changes in the properties of PCa cells and immune cells seem to be important during cancer progression. It is evident that PCa cells can influence lipid metabolism in immune cells in the microenvironment of the tumor and vice versa since both cell types are susceptible to changes in the regulation of lipid metabolism. Furthermore, serum lipid profiling might serve as a target for diagnostic tools and surveillance of PCa progression. 

Thus, it is important to characterize the impact of changes in the regulation of lipid metabolism leading to disturbances in the properties of prostate cancer cells and immune cells in the tumor microenvironment during tumor progression. By clarifying these alterations, we can gain a better understanding of the interaction between PCa and immune responses, thus creating an opportunity to improve the treatment of this life-threatening disease. 

## Figures and Tables

**Figure 1 cancers-14-04293-f001:**
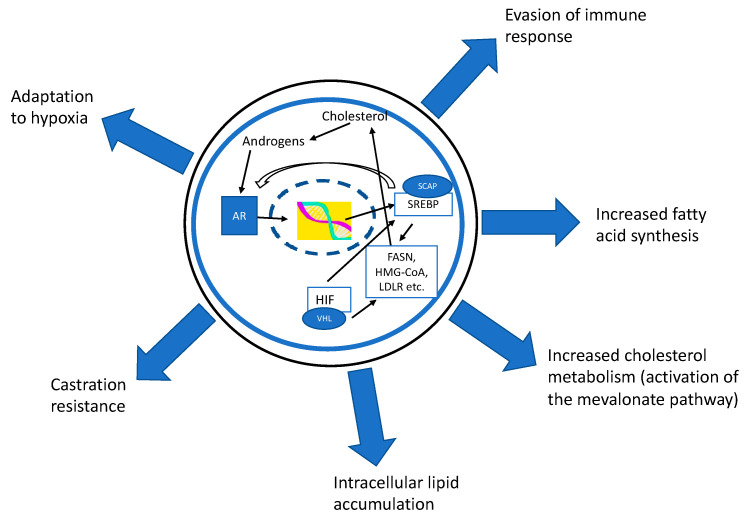
The androgen receptor (AR) modulates the expression of many important genes in the lipid metabolism pathways. Especially sterol regulatory element-binding proteins (SREBP) and fatty acid synthase (FASN) are upregulated in an AR-dependent manner. The AR activates the expression of the SREBP cleavage-activating protein (SCAP) which is needed for SREBP activation. SREBP can induce the activation of FASN, but also upregulates cholesterol metabolism by inducing the mevalonate pathway. SREBP also acts as an oxygen sensor (via hypoxia inducible factor (HIF)) which helps cells to adapt to a hypoxic microenvironment. SREBP increases the expression of AR; thus, the AR–SREBP-axis forms a self-regulating loop ensuring the constant gene expression of transcription factors. The importance of the regulation of lipid metabolism has been demonstrated also in the regulation of tumor-associated immune cells and in development of treatment resistances in prostate cancer.

**Table 1 cancers-14-04293-t001:** Summary of published studies on changes in the lipidome and cholesterol metabolism in cancer-associated immune cells.

Specific Cell Type	Specific Cancer Type (If Known)	Impact of Lipid Metabolism on Immune Cells	Reference
CD8+ T-cells		Cholesterol content in tumor microenvironment increased expression of immune checkpoints leading to T-cell exhaustion. Cholesterol depletion restore the activity	Ma et al., 2019 [87]
Anticancer Tc9 cells (CD8+ T-cells)		Cholesterol depletion increased antitumor activity	Ma et al., 2018 [88]
CD8+ T-cells		Cholesterol in cancer cell’s plasma memrane decrease CD8+ cytotoxicity	Lei et al., 2021 [89]
CD8+ T-cells	melanoma (in a mouse model)	Inhibition of acetyl-CoA cholesterol acyltransferase increased proliferation of CD8+ T-cells	Yang et al., 2016 [90]
CD8+ T-cells	Liver carcinoma cells	Inhibition on cholesterol acyltransferase increased the amount of CD8+ T-cells and restorored CD8+ exhausted T-cell activity	Schmidt et al., 2021 [91]
T-cells	breast cancer, colon cancer, and melanoma	Inhibition of protein convertase subtilisin/kexin 9 decreased tumor growth and mortality in mice by reducing cholesterol metabolism and increasing T-cell infiltration in the tumors	Liu et al., 2020 [92]
Treg cells	melanoma (in a mouse model)	Lipid associated metabolic pathways by SREBP activity were enriched in tumors’ Treg cells. Inhibition of SREBP activity in Treg cells decreased tumor growth	Lim et al., 2021 [83]
T cells and tumor-associated macrophages		Treg cells were able to activate tumor associated macrophages by modulating interferon γ secretion in CD8+ T-cells	Liu et al., 2019 [84]
CD4+ T-cells	Prostate cancer	Condition median from obesity-modified CD4+ T cells (decreased expression of IFNγ and IL-2 and increased expression of IL-6) increased the expression of epithelial-mesenchymal transition markers and showed a higher invasive and migratory capacity	De Angulo et al., 2022 [94]
Macrophages	Prostate cancer	Macrophages were associated with cholesterol transport and androgen synthesis in prostate cancer cells	El-Kenawi et al., 2021 [95]
Macrophages’ polarization		In ABCG1 (transporter which efflux cholesterol from the cells) knockdown mouse strain, macrophage polarization switch from tumor-promoting M2 to anti-tumor-promoting M1 phenotype	Sag et al., 2015 [96]
Macrophages’ polarization		Inhibition of ABCG1 in human macrophages switch polarization from tumor-promoting M2 to anti-tumor-promoting M1 phenotype. Stimulation with HDL also downregulated polarization to M1 phenotype	Lee et al., 2016 [97]
Macrophages	Prostate cancer	Cancer cell derived IL-1β enhanced expression of scavenger receptor, marco, on subset of macrophages. This was associated with prostate cancer progression and shorter disease-free survival. Marco was shown to regulate accumulation of lipids into the macrophages	Masetti et al., 2022 [98]
Myeloid-derived suppressor cells and macrophages	Prostate cancer	In mice fed with high-fat diet, number of myeloid-derived suppressor cells and ratio of M2/M1 macrophages were increased	Hayashi et al., 2018 [99]

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
