# Peer review of "Role of Lipids and Lipid Metabolism in Prostate Cancer Progression and the Tumor’s Immune Environment"

_cancers, 2022, doi:10.3390/cancers14174293_

Round 1

Reviewer 1 Report

The article is interesting, but significant adjustments mentioned below are recommended to be accepted for publication.

The plagiarism of the manuscript is 21%, and for standard journals, the plagiarism should be below 15%. This is a minor concern that needs to be adequately addressed and rectified.

·       What new has been added in this manuscript in comparison to the plethora of literature available related/similar to the subject?

·       In the past few years, the incidence and mortality associated with PCa have gradually declined; however, it is still one of the leading causes of death. You have not mentioned anything related to the global incidence or prevalence in the last decade in your review. In this regard, you can add a table or figure.

·       In the introduction, add some information related to global prevalence and the number of cases of PCa.

·       Mention reference to line 36 in the introductory part.

·       Further, is the occurrence and development of prostate cancer affected by race, family history, microenvironment, and other factors. You can add a few lines related to this with suitable references.

·       In lines 56-57 of the introduction, you mentioned bacterial infection is associated with an increased risk of PCa. Is there any role of prostrate microflora or their metabolites, particularly lipids, in the occurrence, development, and prognosis of PCa?

   Lipid metabolism is regulated by SREBP, which is controlled by multiple mechanisms at levels of mRNA synthesis or transcriptional activity. Can SREBP pathway modulation contribute to the development of PCa.

·       A section related to SREBS’ could be added to the paper.

·       In lines 128-132 please provide a reference

·       In lines 165-168 please provide a reference

·       In lines 172-173, you have mentioned Cholesterol esters were also associated with progression and metastatic disease. Why cholesterol esters are fueling the fury of PCa. Has any study mentioned the same?

·       The heading mentions Lipidome as a biomarker in prostate cancer. How can you justify this statement despite the recent advances in molecular techniques and computational approaches? On what criteria you have selected liposome as a biomarker for PCa and not other metabolites of other pathways that are significantly essential.

·       Why your article has focussed on liposome as a biomarker and not on metabolome or proteome that too could serve as biomarkers of PCa.

      The lipidomic approach will provide insights into the specific function of lipid species in Pca and identify potential biomarkers for establishing preventive or therapeutic programs.

·       How will you justify lipidomics can contribute to understanding the biological mechanisms inherent to PCa and why lipids could be relevant biomarkers for PCa.

·       Is there any role of lipid metabolism in the metabolic reprogramming of macrophages that can lead to the failure of anticancer therapies?

·       In the manuscript, there is only 1 figure and 1 table; a suggestion is that you could add more figures and tables to the review.  

·       A section related to potential molecular targets and their inhibitors for PCa treatment could be added to the review.

·       A section related to critical drives of alterations in lipid metabolism could be added.

·       Analytical and computational approaches that could justify liposome as a biomarker for PCa should be added.

·    Recent advances in managing PCa treatment and drug discovery could be added.

Overall, the manuscript needs restructuring as the quality and content are not enough to be accepted for publication; however, if the suggested changes are added, it should be considered after evaluation. 

Author Response

The article is interesting, but significant adjustments mentioned below are recommended to be accepted for publication.

Comment: The plagiarism of the manuscript is 21%, and for standard journals, the plagiarism should be below 15%. This is a minor concern that needs to be adequately addressed and rectified.

Response: We are grateful for reviewers attention. As many different authors wrote and modified our manuscript, in the end it was impossible to track which part was authors’ own text and which was more or less similar to the references. We have now extensively revised the text during a language check by native English speaker and the plagiarism rate should thus be lower.

Comment: What new has been added in this manuscript in comparison to the plethora of literature available related/similar to the subject?

Response: As far as we know, this is the first review article which covers current knowledge about what is known possible association of lipid metabolism and immune responses in PCa. In general, this is relatively new topic in the field of cancer and especially in the field of PCa research. Thus, in our opinion our review is topical and adds to the current literature.

Comment: In the past few years, the incidence and mortality associated with PCa have gradually declined;

however, it is still one of the leading causes of death. You have not mentioned anything related to the global incidence or prevalence in the last decade in your review. In this regard, you can add a table or figure.

Response: We have added now sentence about this topic to the introduction part. As epidemiology of prostate cancer is not the core focus of this review article, we have not increased the number of tables or figures, which we feel should be focused on the core topics. (lines: 38-43)

Comment: In the introduction, add some information related to global prevalence and the number of cases of PCa.

Response: We have now added more specific information about incidence and mortality rates in the introduction part of our manuscript. (lines: 38-43)

Comment: Mention reference to line 36 in the introductory part.

Response: We have now added reference to this part.

Comment: Further, is the occurrence and development of prostate cancer affected by race, family history, microenvironment, and other factors. You can add a few lines related to this with suitable references.

Response: We have now included more text on this topic into beginning of the introduction part. However, as this is not a epidemiological review, we keep this part compact. (lines: 38-43)

Comment: In lines 56-57 of the introduction, you mentioned bacterial infection is associated with an increased risk of PCa. Is there any role of prostrate microflora or their metabolites, particularly lipids, in the occurrence, development, and prognosis of PCa?

Response: This is interesting point, however, research related on direct and indirect impact of human microbiota and PCa is sparse and the topic is not well understood. We added one sentence about the topic into the introduction part with reference recent review article, thus, readers interested on the topic can have further look on it. (lines: 61-64).

Comment: Lipid metabolism is regulated by SREBP, which is controlled by multiple mechanisms at levels of mRNA synthesis or transcriptional activity. Can SREBP pathway modulation contribute to the development of PCa.

Response: Most probably SREBP mediated pathways play a role during PCa development. However, the

importance of SREBP-mediated lipid metabolism is probably greater during disease progression. As PCa is dependent on androgen signalling and SREBP can affect androgen receptor activation and vice versa, it is likely that these pathways also contribute to modulation during development of PCa.

Comment: A section related to SREBS’ could be added to the paper.

Response: We agree, as SREBPs have crucial role in regulation of lipid metabolism we have now added a new paragraph devoted to SREBPs. (lines: 119-129)

Comment: In lines 128-132 please provide a reference.

Response: We are grateful for reviewers accuracy and we have provided reference to this sentence.

Comment: In lines 165-168 please provide a reference

Response: We have now add reference to this sentence.

Comment: In lines 172-173, you have mentioned Cholesterol esters were also associated with progression and metastatic disease. Why cholesterol esters are fueling the fury of PCa. Has any study mentioned the same?

Response: Cholesterol esters are the form in which cells are storing inner cell cholesterol as free intracellular cholesterol itself is toxic to the cells. Therefore all studies in our review that have explored role of intracellular cholesterol have basically studied cholesterol esters. Thus, this notion is in line with the published studies which shows importance of cholesterol, cholesterol metabolites, and cholesterol metabolism during PCa progression.

Comment: The heading mentions Lipidome as a biomarker in prostate cancer. How can you justify this statement despite the recent advances in molecular techniques and computational approaches? On what criteria you have selected liposome as a biomarker for PCa and not other metabolites of other pathways that are significantly essential.

Response: We agree that many other metabolites and pathways are under investigation to be a new marker for PCa detection and surveillance. However, as our current topic is lipids and lipid metabolism in PCa, we limited our discussion only on lipidome. Other types of possible biomarkers are the topic of other review articles.

Comment: Why your article has focussed on liposome as a biomarker and not on metabolome or proteome that too could serve as biomarkers of PCa.

Response: The topic of our review is to cover current knowledge about lipids, lipid metabolism, immune responses and prostate cancer. As mentioned above, reviewing the field of other new biomarkers in PCa is beyond the scope of this article.

Comment: The lipidomic approach will provide insights into the specific function of lipid species in Pca and identify potential biomarkers for establishing preventive or therapeutic programs.

Response: We agree that this is an exciting and promising topic. However, more research with larger data will be needed before any final conclusions.

Comment: How will you justify lipidomics can contribute to understanding the biological mechanisms inherent to PCa and why lipids could be relevant biomarkers for PCa.

Response: This topic is still under evaluation, however, lipidomic is quite stable and easy to measure from blood samples. As we show in our review, there is accumulating evidence on important role of lipidomics in prostate cancer progression, which justifies finding out whether lipidomics could also serve as biomarker.

Comment: Is there any role of lipid metabolism in the metabolic reprogramming of macrophages that can lead to the failure of anticancer therapies?

Response: The possible link between lipid metabolism and immune responses are so called ‘hot topic’ at the field now as IO treatments more or less failed in PCa treatment. Thus, we already know that immune cells infiltrating the prostate do not function as actively after IO activation as in some other solid cancer types. One explanation to this might be modulation of lipid metabolism, which may affect immune cell activity. However, this is still under investigation. Role of metabolic reprogramming of immune cells in treatment resistance to other types of anticancer treatments apart from IO therapy is currently unstudied area.

Comment: In the manuscript, there is only 1 figure and 1 table; a suggestion is that you could add more figures and tables to the review.  

Response: The journal’s Instructions to authors required two figures and/or tables, thus, we have met the requirement.

Comment: A section related to potential molecular targets and their inhibitors for PCa treatment could be added to the review.

Response: We agree that this is interesting and important point, however, this topic is too wide and beyond the scope of this review. Once again this could be interesting topic on review on its own. However, we do discuss FAS and SREBP as potential novel targets for PCa treatment. 

Comment: A section related to critical drives of alterations in lipid metabolism could be added.

Response: We agree that this is important aspect. We have covered in detail functions of SREBP, FAS and the mevalonate pathway which are most important drivers of lipid metabolism and it’s alterations. However, the topic of covering all external influences that might alter lipid metabolism is too wide for this review as there are so many disorders and stages in human body which might drive the alteration of lipid metabolism such as cardiovascular diseases and events, diabetes mellitus, diet in general, metabolic syndrome and other life-style related stages. Here is a recent review article which covers at least partly this topic: Natesan & Kim. Lipid Metabolism, Disorders and Therapeutic Drugs - Review. Biomol Ther (Seoul) 2021;29:596-604. We have added this reference to the manuscript. (lines: 130-132)

Comment: Analytical and computational approaches that could justify liposome as a biomarker for PCa should be added.

Response:We agree that analysis of liposome as biomarker requires advanced statistical methodology. Again, this topic is too wide for this review, extensive discussion would blur the focus. Our groups has approached this issue recently in a thesis: https://aaltodoc.aalto.fi/handle/123456789/113931

Comment: Recent advances in managing PCa treatment and drug discovery could be added.

Response: PCa treatment, especially treatment of castration resistant and advcanced prostate cancer has indeed improved vastly in recent decade. We are currently in an era when personalized treatment decisions are possible also in prostate cancer. However, we have to limit out topic, thus we have to leave this section out of the current manuscript. This topic have been comprehensively reviewed elsewhere (see Nevedomskaya et al. Int J Mol Sci 2018, Yap et al. Nature Reviews Drug Discovery 2016, Sayegh et al. JCO Oncol Pract 2022).

Overall, the manuscript needs restructuring as the quality and content are not enough to be accepted for publication; however, if the suggested changes are added, it should be considered after evaluation.

Response: We are thankful for the comments. It is important to keep the manuscript focused, therefore we have not extensively included all of the points discussed here. However, we have revised the structure of the manuscript according to reviewer comments, and the language has been checked by a native English speaker. We hope the revised manuscript will be suitable for publication.

Reviewer 2 Report

Siltari et al present a review about role and associations between lipids and lipid metabolism on prostate cancer. They further explore the role of lipid metabolism in the interaction between cancer cells and immune cells. The authors superficially cover the role of SREBPs, altered lipid compositions in prostate cancer, using lipids as biomarkers, cholesterol metabolism, fatty acid metabolism, potential lipid-related drug targets and lipid metabolism of immune cells and their effect on cancer development.

It is a very clearly written and interesting manuscript on a highly intriguing topic. This is one of the manuscripts that I am glad I accepted being a reviewer on (which is not always the case). However, I do have several points that needs to be addressed before publication:

·         The description of the prostate cancer prevention trial (lines 50-55) is a bit too specific to fit well in the introduction. It would be better to move it to one of the specific sections

·         Paragraph on lines 58-64 should have references

·         Better description of SREBPs: Considering a whole section is given to the role of SREBPs, there is a lack of background regarding what SFRBPs actually do. Yes, they are modulators, they induce certain lipid pathways etc. But how do they work? Are they transcription factors? Do they directly activate synthesis proteins?

·         There seems to be a shift in the references, citation 15 (in the text) should be 14 and so on.

·         First paragraph of cholesterol metabolism section (lines 229-238): first of all, lack citations. And secondly, is slightly misleading. The reader is left with the impression that cholesterol synthesis is unique to prostate cancer development. In actuality the epithelial cells of the normal prostate has a high production rate and high levels of cholesterol, and may even have as high synthesis as the lever [1]. These high cholesterol levels appear to increase even further with cancer progression. Although cell culture studies imply this is due to increased de novo synthesis of cholesterol, analysis of human prostate cancer tissue is more unclear. It may be higher synthesis, or higher cholesterol uptake from the blood.

·         A proper presentation and discussion on using lipid and fatty acids for energy production through beta-oxidation and the carnitine shuttle is missing. The authors slightly touch on these mechanisms here and there (lines, 368), but it not fully explained or clear how utilization of lipids for energy production changes in prostate cancer progression. Suggestions for literature to include and check out: [2-5]

·         For the topic of this review, it is really worth to include a paragraph or so on the role of periprostatic adipose tissue (PPAT) [6]. There are indications that this adjacent fat tissue feed the prostate with lipids, pro-inflammatory cytokines and chemokines.

·                     1.            Freeman, M.R. and K.R. Solomon, Cholesterol and benign prostate disease. Differentiation, 2011. 82(4-5): p. 244-52.

·                     2.            Valentino, A., et al., Deregulation of MicroRNAs mediated control of carnitine cycle in prostate cancer: molecular basis and pathophysiological consequences. Oncogene, 2017. 36(43): p. 6030-6040.

·                     3.            Bramhecha, Y.M., et al., Fatty acid oxidation enzyme Δ3, Δ2-enoyl-CoA isomerase 1 (ECI1) drives aggressive tumor phenotype and predicts poor clinical outcome in prostate cancer patients. Oncogene, 2022. 41(20): p. 2798-2810.

·                     4.            Andersen, M.K., et al., Spatial differentiation of metabolism in prostate cancer tissue by MALDI-TOF MSI. Cancer & Metabolism, 2021. 9(1): p. 9.

·                     5.            Qu, Q., et al., Fatty acid oxidation and carnitine palmitoyltransferase I: emerging therapeutic targets in cancer. Cell Death Dis, 2016. 7: p. e2226.

·                     6.            Estève, D., et al., Periprostatic adipose tissue: A heavy player in prostate cancer progression. Current Opinion in Endocrine and Metabolic Research, 2020. 10: p. 29-35.

Author Response

Siltari et al present a review about role and associations between lipids and lipid metabolism on prostate cancer. They further explore the role of lipid metabolism in the interaction between cancer cells and immune cells. The authors superficially cover the role of SREBPs, altered lipid compositions in prostate cancer, using lipids as biomarkers, cholesterol metabolism, fatty acid metabolism, potential lipid-related drug targets and lipid metabolism of immune cells and their effect on cancer development.

It is a very clearly written and interesting manuscript on a highly intriguing topic. This is one of the manuscripts that I am glad I accepted being a reviewer on (which is not always the case). However, I do have several points that needs to be addressed before publication:

Response: Thank you for the supportive feedback. Your comments and suggested improvements were very helpful and made our review even better! All addition in the revised manuscript have been marked with red colored text.

Comment:  The description of the prostate cancer prevention trial (lines 50-55) is a bit too specific to fit well in the introduction. It would be better to move it to one of the specific sections

Response: We agree and have moved the PCPT description to the section of Pharmacological interventions targeting lipid metabolism affect immune cells (lines: 569-574).

Comment: Paragraph on lines 58-64 should have references

Response: We have now added reference to this paragraph.

Comment: Better description of SREBPs: Considering a whole section is given to the role of SREBPs, there is a lack of background regarding what SFRBPs actually do. Yes, they are modulators, they induce certain lipid pathways etc. But how do they work? Are they transcription factors? Do they directly activate synthesis proteins?

Response: We agree, as SREBPs have crucial role in regulation of lipid metabolism we have now added a new paragraph devoted to background of SREBPs (lines:119-129).

Comment: There seems to be a shift in the references, citation 15 (in the text) should be 14 and so on.

Response: We are grateful for reviewer’s accuracy. Listing of the references are now checked and corrected.

Comment: First paragraph of cholesterol metabolism section (lines 229-238): first of all, lack citations. And secondly, is slightly misleading. The reader is left with the impression that cholesterol synthesis is unique to prostate cancer development. In actuality the epithelial cells of the normal prostate has a high production rate and high levels of cholesterol, and may even have as high synthesis as the lever [1]. These high cholesterol levels appear to increase even further with cancer progression. Although cell culture studies imply this is due to increased de novo synthesis of cholesterol, analysis of human prostate cancer tissue is more unclear. It may be higher synthesis, or higher cholesterol uptake from the blood.

Response: We agree, We have now clarified this paragraph accordingly and added the references (lines:267-271).

Comment: A proper presentation and discussion on using lipid and fatty acids for energy production through beta-oxidation and the carnitine shuttle is missing. The authors slightly touch on these mechanisms here and there (lines, 368), but it not fully explained or clear how utilization of lipids for energy production changes in prostate cancer progression. Suggestions for literature to include and check out: [2-5]

Response: We agree that this aspect of fatty acids has been missing in our review. Thus, we have now added paragraph discussing how fatty acids’ beta-oxidation and carnitine shuttle is upregulated in PCa based on the references which reviewer kindly provided. Thank you! (lines: 414-425)

Comment: For the topic of this review, it is really worth to include a paragraph or so on the role of periprostatic adipose tissue (PPAT) [6]. There are indications that this adjacent fat tissue feed the prostate with lipids, pro-inflammatory cytokines and chemokines.

Response: Thank you for providing this interesting point for us! We have now added a new chapter about the possible role of PPAT in PCa at the end of our manuscript (lines: 606-620).

  • 1.            Freeman, M.R. and K.R. Solomon, Cholesterol and benign prostate disease.Differentiation, 2011. 82(4-5): p. 244-52.
  • 2.            Valentino, A., et al., Deregulation of MicroRNAs mediated control of carnitine cycle in prostate cancer: molecular basis and pathophysiological consequences.Oncogene, 2017. 36(43): p. 6030-6040.
  • 3.            Bramhecha, Y.M., et al., Fatty acid oxidation enzyme Δ3, Δ2-enoyl-CoA isomerase 1 (ECI1) drives aggressive tumor phenotype and predicts poor clinical outcome in prostate cancer patients.Oncogene, 2022. 41(20): p. 2798-2810.
  • 4.            Andersen, M.K., et al., Spatial differentiation of metabolism in prostate cancer tissue by MALDI-TOF MSI.Cancer & Metabolism, 2021. 9(1): p. 9.
  • 5.            Qu, Q., et al., Fatty acid oxidation and carnitine palmitoyltransferase I: emerging therapeutic targets in cancer.Cell Death Dis, 2016. 7: p. e2226.
  • 6.            Estève, D., et al., Periprostatic adipose tissue: A heavy player in prostate cancer progression.Current Opinion in Endocrine and Metabolic Research, 2020. 10: p. 29-35.